# Aberrant DNA and RNA Methylation Occur in Spinal Cord and Skeletal Muscle of Human SOD1 Mouse Models of ALS and in Human ALS: Targeting DNA Methylation Is Therapeutic

**DOI:** 10.3390/cells11213448

**Published:** 2022-10-31

**Authors:** Lee J. Martin, Danya A. Adams, Mark V. Niedzwiecki, Margaret Wong

**Affiliations:** 1Department of Pathology, Division of Neuropathology, Johns Hopkins University School of Medicine, Baltimore, MD 21205, USA; 2Department of Neuroscience, Johns Hopkins University School of Medicine, Baltimore, MD 21205, USA; 3Department of Anesthesiology and Critical Care Medicine, Johns Hopkins University School of Medicine, Baltimore, MD 21205, USA; 4Pathobiology Graduate Training Program, Johns Hopkins University School of Medicine, Baltimore, MD 21205, USA

**Keywords:** motor neuron, cytosine methylation, 6-methyladenosine, FTO, Dnmt3A, CpG island

## Abstract

Amyotrophic lateral sclerosis (ALS) is a fatal disease. Skeletal muscles and motor neurons (MNs) degenerate. ALS is a complex disease involving many genes in multiple tissues, the environment, cellular metabolism, and lifestyles. We hypothesized that epigenetic anomalies in DNA and RNA occur in ALS and examined this idea in: (1) mouse models of ALS, (2) human ALS, and (3) mouse ALS with therapeutic targeting of DNA methylation. Human superoxide dismutase-1 (hSOD1) transgenic (tg) mice were used. They expressed nonconditionally wildtype (WT) and the G93A and G37R mutant variants or skeletal muscle-restricted WT and G93A and G37R mutated forms. Age-matched non-tg mice were controls. hSOD1 mutant mice had increased DNA methyltransferase enzyme activity in spinal cord and skeletal muscle and increased 5-methylcytosine (5mC) levels. Genome-wide promoter CpG DNA methylation profiling in skeletal muscle of ALS mice identified hypermethylation notably in cytoskeletal genes. 5mC accumulated in spinal cord MNs and skeletal muscle satellite cells in mice. Significant increases in DNA methyltransferase-1 (DNMT1) and DNA methyltransferase-3A (DNMT3A) levels occurred in spinal cord nuclear and chromatin bound extracts of the different hSOD1 mouse lines. Mutant hSOD1 interacted with DNMT3A in skeletal muscle. 6-methyladenosine (6mA) RNA methylation was markedly increased or decreased in mouse spinal cord depending on hSOD1-G93A model, while fat mass and obesity associated protein was depleted and methyltransferase-like protein 3 was increased in spinal cord and skeletal muscle. Human ALS spinal cord had increased numbers of MNs and interneurons with nuclear 5mC, motor cortex had increased 5mC-positive neurons, while 6mA was severely depleted. Treatment of hSOD1-G93A mice with DNMT inhibitor improved motor function and extended lifespan by 25%. We conclude that DNA and RNA epigenetic anomalies are prominent in mouse and human ALS and are potentially targetable for disease-modifying therapeutics.

## 1. Introduction

ALS is a fatal neurological disease characterized by initial muscle spasticity and fasciculation, followed by muscle weakness, atrophy, and eventual paralysis and respiratory arrest [1,2]. The cause of the muscle spasticity, paralysis, and atrophy and the body mass wasting is believed to be neuromuscular junction deterioration and progressive degeneration and elimination of upper motor neurons (MNs) in cerebral cortex and lower MNs in brainstem and spinal cord [1,2,3]. Patients die typically 3 to 5 years after diagnosis. More than 5000 people in the USA and nearly 120,000 individuals worldwide are diagnosed with ALS each year (https://alsnewstoday.com/als-facts-statistics, accessed 29 October 2022). Aging is a strong risk factor [4]; ALS onset commonly occurs between 40 to 70 years of age. Other known risk factors are maleness and family history [5,6].

The molecular pathogenesis of ALS is understood poorly [6,7], contributing to the lack of comprehensive tissue and molecular target identification; thus, effective mechanism-based, disease-modifying therapies are lacking [2]. Most individuals who develop classical ALS are known as sporadic with few known genetic contributions [1,6]. Familial forms of ALS are autosomal dominant, or autosomal recessive, and make up ~10% of all ALS cases. ALS-linked gene mutations occur in *sod1*, *tdp43*, *fus*, and *c9orf72* [6]. Sporadic forms are clinically and neuropathologically indistinguishable from familial forms [1,3,7], raising the possibility that the disease is modified epigenetically. Disease discordance in monozygotic twins and triplets supports this possibility [8].

An epigenetic modification to nucleic acids can mimic genetic change and anomaly where Mendelian inheritance is lacking [9,10]. Epigenetic modifications to DNA are reversible, heritable, and non-heritable changes in gene function that occur without a change in DNA sequence [9,10]. 5-methylcytosine (5mC) methylation occurs in cytosine-phosphate-guanine (CpG) dinucleotides within the 5′ regulatory regions of genes [11,12]. Cytosine methylation modulates gene transcription, imprinting, and X-chromosome inactivation and is thought to participate in normal development and several aberrant disease processes in vertebrates [13,14,15,16]. There is accruing interest in DNA methylation in ALS [17,18]. DNA methylation is mediated enzymatically by DNA methyltransferase (DNMT) [19,20,21,22]. DNMT1 is an abundant isoform in proliferating cells and displays a preference for hemi-methylated substrates and is targeted to replication forks, acting to maintain methylation patterns during cell replication [23] and to DNA repair sites [24]. Mutations in the *dnmt1* gene can cause neurodegenerative disease, including hereditary sensory and autonomic neuropathy type 1 [25] and cerebellar ataxia [26]. DNMT3A and DNMT3B function as de novo methyltransferases because they methylate hemi-methylated DNA and completely unmethylated DNA [27,28]. DNMT3A is abundant in nervous and skeletal muscle tissue, but DNMT3B is not [29,30]. Mutations in the *dnmt3a* gene cause overgrowth syndrome with macroencephaly, intellectual disability, and autism spectrum disorder [31]. DNMT1 has been found in mitochondria from cultured mouse embryonic fibroblasts [32]. DNMT3A has been found in pure mitochondria from human brain and mouse skeletal muscle and CNS; it is enriched in certain tissues and matches the content of 5mC in mitochondrial DNA [30].

Epigenetic modification of RNA also occurs [33], though there is debate about its relevance and function [34]. RNA-base post-transcriptional methylations are in messenger RNA (mRNA), transfer RNA (tRNA), ribosomal RNA, and small nuclear RNA [33]. N^6^-methyladenosine (6mA) is a common reversible mRNA modification with putative roles in cell differentiation, cancer, adipogenesis, lipid biology, and inflammation, but functions of 6mA in the nervous system are unclear. The anticodon loop of tRNA is methylated by DNMT2, also known as tRNA aspartic acid methyltransferase-1 (TRDMT1) [35]. Other writers of RNA methylation are methyltransferase-like protein 3 (METTL3), METTL14, METTL16, and Wilms tumor 1-associated protein [36,37,38]. Alk B homolog 5 (ALKBH5) and fat mass and obesity associated protein (FTO) are RNA demethylases—the erasers [36]. Additional proteins, the readers, interact with 6mA-methylated RNA [36]. RNA methylation regulates RNA trafficking, stability, and degradation [39,40]. 6mA mRNA modifications are written in nascent pre-mRNA to specify nuclear export and cytoplasmic turnover [41]. Emerging data suggest that 6mA has a role in metabolic diseases and RNA disorders [36,37,38].

Work on nucleic acid epigenetics is pertinent to ALS because aberrant DNA methylation could participate in human ALS pathogenesis [18,29,30,42]. DNMT1 and DNMT3A proteins are upregulated in human ALS MNs [29]. In in vivo experimental neuronal cell death settings, spinal MNs can engage epigenetic mechanisms to drive cell death, involving DNMT upregulation, increased enzymatic activity, and increased DNA methylation [29]. Indeed, a small molecule inhibitor of DNMTs, RG108, blocked the elevations in DNMT enzymatic activity, 5mC accumulation, and strongly protected spinal MNs from degeneration in vivo [29]. Human superoxide dismutase-1 (hSOD1) transgenic (tg) mouse skeletal muscle and CNS have abnormal levels of mitochondrial DNA methylation [30]. Primary disease mechanisms for ALS could be engaged in skeletal muscle [43,44,45]. Skeletal muscle and adipose tissue involvement in human ALS and mouse models of ALS [43,44,45,46] suggests that the ALS syndrome is a metabolic disease in human [47] and mouse models [48,49]. This idea dovetails with functions of RNA epigenetics as a regulator of metabolic disease [36]. RNA dysfunction has been implicated in ALS pathogenesis [50,51,52,53]. This study was undertaken to test three hypotheses: (1) DNA and RNA epigenetic markers and epigenetic regulatory mechanisms are aberrant in skeletal muscle and spinal cord of hSOD1 mouse models of ALS; (2) epigenetic anomalies occur in human ALS postmortem brain and spinal cord; and (3) DNA methylation is therapeutically targetable and disease-modifying in mice, thus implicating epigenetics as a mechanism of disease in ALS. 

## 2. Materials and Methods

### 2.1. Transgenic (tg) Mice

We used several different lines of tg mice with *hsod1* as mutant or wildtype (WT) alleles. The mouse lines expressed hSOD1 nonconditionally (throughout the whole body) or conditionally in skeletal muscle (hSOD1^mus^) only. We used these different tg mouse lines to compare potential epigenetic pathology in the global presence of disease-causing genes and in the tissue-confined presence of disease-causing genes in skeletal muscle to identify CNS non-autonomous epigenetic pathology. All mouse lines have normal endogenous mouse *sod1* alleles. Non-tg age- and sex-matched littermates were controls. All mice were barcoded and genotyped from tail genomic DNA at 1 month of age. With all tg mice, when limb weakness developed, ad libitum NutriGel and chow pieces were placed in the cage, and water was available at a bottom-cage level drinking spout. The institutional Animal Care and Use Committee approved the animal protocols. 

The different nonconditional tg mouse lines expressed the Gly93 to Ala substitution (G93A) mutation, the Gly37 to Arg substitution (G37R) mutation, or WT hSOD1. The transgenes are driven by the endogenous hSOD1 promoter. The lines were originally purchased from The Jackson Laboratory (Bar Harbor, ME): G1H line, B6SJL-TgN-SOD1-G93A^1Gur^ [54], B6.Cg-Tg SOD1-G37R 29Dpr/J (line 29), and B6SJL-TgN[SOD1]^2Gur^ [54]. These mouse lines have been used widely by us [55,56,57,58,59] and others as the major therapeutic testing animal model of ALS [60]. 

The hSOD1-G93A mice express a high copy number of human mutant allele (~20 copies) and have a rapid disease course with terminal stages of paralysis at 16–20 weeks of age. The disease in these mice is fully penetrant with presymptomatic stages (<10 weeks of age), early symptomatic stages (10–15 weeks of age) as indicated by leg tremor and paresis and nascent motor deficits, and terminal stages (16–20 weeks of age) as seen by body mass wasting and paralysis. The hSOD1-G37R mice have ~7-fold increases in SOD1 activity. Disease onset (mean ± standard deviation [SD]), as assessed by running wheel activity, and hind-limb paresis, occur at approximately 159 ± 39 days [57]. Median lifespan (mean ± SD) is approximately 366 ± 29 days [57]. The hSOD1-WT mice express levels of normal hSOD1 comparable to the hSOD1-G93A line. These mice do not develop the aggressive disease as their G93A counterparts though they do form cellular inclusions and late onset mitochondriopathy in spinal cord [54,55,61].

We also used tg mice expressing WT-, G37R-, and G93A-hSOD1 only in skeletal muscle. These mice have been described [30,44,45]. They express hSOD1 transgene variants at ~50–110% of endogenous mouse SOD1 in striated skeletal muscles, including diaphragm [44,45]. hSOD1 is not detected in any spinal cord or brain neural cells and other organ systems and tissue types [30,44]. hSOD1^mus^ mice at 1.5 years show significant deficits in motor function detected in holding/hanging, running, and swimming tests [44,45]. Affected mice show progressive limb weakness and paresis with motor deficits, paralysis, and shortened lifespan. All hSOD1^mus^ tg mice become clinical with weakness and limb paresis. Lifespan is significantly shortened by 10–16%. 

### 2.2. Human CNS Tissues 

Postmortem human brain and spinal cord samples were obtained from the Human Brain Resource Center in the Department of Pathology, Division of Neuropathology, Johns Hopkins University School of Medicine. Control (*n* = 12) and sporadic ALS (*n* = 12) were matched for age and postmortem delay [62,63,64]. Brains and spinal cords were immersion fixed in 10% neutral-buffered formalin according to standard autopsy procedures. After fixation, samples of motor cortex (Brodmann area 4) and spinal cord (cervical and lumbar segments) were dissected and paraffin processed for neuropathological final diagnosis and immunohistochemistry [62].

### 2.3. DNMT Enzyme Activity

Mice received a lethal dose of anesthetic and were decapitated for harvesting tissues that were snap-frozen in isopentane cooled by dry ice. Total DNMT enzymatic activity (maintenance and de novo) in spinal cord and skeletal muscle tissue extracts was determined using a validated colorimetric ELISA-like assay (Epigentek, Farmingdale, NY, USA) as described [29].

### 2.4. 5mC and 6mA Assays 

Mouse tissues that were isolated, snap-frozen, and stored at –70 °C were used to extract with Trizol reagent (ThermoFisher Scientific, Waltham, MA, USA) genomic DNA and total RNA from selected mouse skeletal muscles and spinal cord. Postmortem human spinal cords from ALS patients and age-matched control individuals were also used for 6mA assays. ELISA was used to determine 5mC and 6mA concentrations using validated colorimetric ELISA assays (Epigentek). 

### 2.5. Genome-Wide CpG DNA Methylation Profiling in Skeletal Muscle of ALS Mice by Methylated DNA-Immunoprecipitation (MeDIP)-Chip Analysis

Genomic DNA was isolated from snap-frozen hindleg skeletal muscle of low-copy presymptomatic hSOD1^mus^-G37R mice (*n* = 3) and age/gender-matched littermate controls (*n* = 3) and used for CpG island microarrays. All MeDIP, sample labeling, hybridization, and processing were performed under standardized and strict operating procedures. Isolated genomic DNA was sonicated to generate random fragments of 200–1000 bp. Immunoprecipitation (IP) of methylated DNA was performed using BiomagTM magnetic beads coupled to monoclonal antibody to 5mC. The total input and matched IP DNA were labeled with Cy3- and Cy5-labeled random 9-mers, respectively, and hybridized to NimbleGen RN34 Meth 3x720K CpG plus Promoter arrays. An Axon GenePix 4000B microarray scanner was used. Raw data was extracted as pair files by NimbleScan software (Roche NimbleGen Inc., Basal, Switzerland). Median-centering quartile normalization and linear smoothing by Ringo, limma, and MEDME was performed. From the normalized log2-ratio data, a sliding-window peak-finding algorithm as part of NimbleScan v2.5 (Roche NimbleGen Inc.) was applied to find the methylated CpG enriched peaks with specified parameters (sliding window width: 750 bp; mini probes per peak: 2; *p*-value minimum cut-off: 2; maximum spacing between nearby probes within peak: 500 bp). To compare promoters with differentially enriched methylation between hSOD1^mus^-G37R and control mice, log2-ratio values were averaged and then used to calculate the M’ value [M’ = Average(log2 MeDIPE/InputE) − Average(log2 MeDIPC/InputC)] for each probe. NimbleScan sliding-window peak-finding algorithm was run on this data to find the differential enrichment peaks. The differential enrichment peaks were filtered according to the following criteria: (1) at least one of the two groups had the median value of log2 MeDIP/Input ≥ 0.3 and a median value of M’ > 0 within each peak region; and (2) at least half of the probes in a peak had the median value of coefficient of variability ≤ 0.8 in both groups within each peak region.

### 2.6. Immunohistochemistry (IHC) and Immunofluorescence (IF)

For histology, mice were perfusion-fixed with 4% paraformaldehyde. After perfusion-fixation, forelimb triceps muscle, hindlimb biceps femoris muscle, and CNS were removed from each mouse, and the tissues were cryoprotected (20% glycerol) before they were frozen-sectioned (40 μm) using a sliding microtome. Serial tissue sections were stored individually in 96-well plates. Mouse skeletal muscle sections were used for IF to study the localizations of hSOD1 and satellite cell markers [65] as described [30]. The primary antibodies used are in Table 1. 

Mouse spinal cord sections and human motor cortex and spinal cord paraffin sections were used for immunoperoxidase IHC with diaminobenzidine (DAB) as chromogen to localize 5mC and 6mA with monoclonal antibodies (Table 1). Human spinal cord sections were also used for dual antigen labeling to localize 5mC within parvalbumin spinal interneurons visualized with DAB and benzidine dihydrochloride, respectively, as chromogens as described [66,67,68]. Counterstaining with cresyl violet was done for cellular and neuroanatomical perspective.

### 2.7. Cell Counting 

Profile counting was used to estimate the numbers of: (1) mouse skeletal muscle satellite cells positive for hSOD1, MyoD, and 5mC; (2) mouse spinal MNs with nuclear 5mC immunoreactivity (IR); (3) human motor cortex layer V neurons and spinal cord ventral horn MNs positive for 5mC and 6mA; (4) human motor cortex subcortical white matter glial cells positive for 6mA; and (5) human spinal cord interneurons positive for 5mC and parvalbumin. For all IHC experiments and cell counting, series of cut sections were selected with a random start and then systematically sampled (every 10th section) to generate a subsample of anatomically-matched sections from each mouse muscle or spinal cord that were processed and mounted on glass slides for evaluation. For human material, paraffin blocks of motor cortex (Brodmann area 4) and cervical and lumbar spinal cord were cut serially, and neuroanatomically matched sections were selected for staining. MNs and glia were counted at 1000× magnification using strict morphological criteria. The criteria for neurons included a multipolar shape and large round or ellipsoid nucleus, globular Nissl staining of the cytoplasm, and a diameter of ~15–25 μm. The criteria for glial cells in white matter included small size (diamater of 10 μm or less) and scant cytoplasm. 

### 2.8. Electron Microscopy (EM)

Age-matched non-tg and hSOD1^mus^-G37R, -G93A, and -WT tg mice (15–17 months of age) received an anesthetic overdose and were perfused transcardially with 2% paraformaldehyde/2% glutaraldehyde. The group sizes were three mice/genotype. Tissue samples of left and right biceps femoris were taken from each mouse and processed and embedded in plastic for conventional transmission EM as described [69]. The sections were viewed and imaged using a Phillips CM12 electron microscope. Digital electron micrographs from each mouse genotype were used to examine satellite cells in at least 20 images per mouse. 

### 2.9. Immunoblotting and IP

Western blotting was done to measure protein levels of DNMT1, DNMT3A, METTL3, ALKBH5, and FTO in mouse skeletal muscle and spinal cord. Age-matched tg and non-tg littermate control mice (*n* = 6/group) received a lethal dose of anesthesia and were decapitated for harvesting forelimb and hindlimb skeletal muscle and spinal cord. The subcellular fractionation protocol has been verified [63,70]. Protein concentrations were measured by a Bio-Rad protein assay with bovine serum albumin as a standard. 

Extracts from skeletal muscle and spinal cord and positive control cell lysates were subjected to sodium dodecyl sulfate polyacrylamide gel electrophoresis (SDS-PAGE) and transferred to nitrocellulose membrane by electroelution as described [70]. For IP, prior to SDS-PAGE, skeletal muscle protein extract (500 µg) was input for 5 µg human-specific hSOD1 antibody (Table 1) followed by agarose-conjugated protein A (ThermoFisher) for capture. Negative control conditions were tissue homogenates immunoprecipitated with isotype specific non-immune IgG, with PBS but no IgG, IP with specific primary antibody but with no homogenate input, and non-tg mouse skeletal muscle extracts [45]. Final IP samples were subjected to SDS-PAGE and Western blot. The reliability of sample loading and electroblotting in each experiment was evaluated by staining nitrocellulose membranes with Ponceau S before immunoblotting. Ponceau S-stained membranes were used in the figures to show protein loading. If transfer was not uniform, blots were discarded, and SDS-PAGE was repeated. Blots were blocked with 2.5% nonfat dry milk with 0.1% Tween 20 in 50 mM Tris-buffered saline (pH 7.4), then incubated overnight at 4 °C with antibody (Table 1). The antibodies were used at concentrations for visualizing protein immunoreactivity (IR) within the linear range. After the primary antibody incubation, blots were washed and incubated with horseradish peroxidase-conjugated secondary antibody (0.2 µg/mL), developed with enhanced chemiluminescence (ThermoFisher), and exposed to X-ray film or imaged with a ChemiDoc imaging system (Bio-Rad Laboratories, Hercules, CA, USA). ImageJ software was used to quantify target protein IR normalized to protein loading identified by Ponceau S staining. 

### 2.10. RG108 Therapeutic Testing in Mouse ALS

To assess a contribution of DNA methylation mechanisms in a mouse model of ALS, nonconditional high-expressing hSOD1-G93A mice were treated daily with 2 mg/kg (intraperitoneal, ip) RG108 (Sigma, Springfield, MO, USA) starting at 6 weeks of age through terminal disease. RG108 is a small, cell-permeable, non-nucleoside direct specific inhibitor of DNMT [71]. Because RG108 does not have a nucleoside derivation such as 5-azacytidine, 5-fluro-2-deoxycytidine, and zebularine, it has low to no toxicity [72]. We verified using histopathological (heart, liver, gut) and behavioral (motor activity) endpoints that RG108 has no apparent toxicity in adult non-tg male mice with chronic systemic treatment at doses of 2 mg/kg and 10 mg/kg or vehicle (DMSO/saline), ip (Appendix A). Afterwards, forty tg male mice were randomized to treatments of RG108 (*n* = 20) or vehicle (*n* = 20). At 10–12-weeks of age, running wheel motor activity was tested on all mice. A subset of mice from each group (*n* = 6) was perfusion-fixed for histology to use for in vivo biological efficacy of drug action as assessed by skeletal muscle satellite cells positive for 5mC. The remaining mice were used for time-to-event outcomes. 

### 2.11. Data Analysis

The values shown in the graphs represent the mean ± SD unless stated otherwise. For histological data, group means and variances were evaluated statistically by one-way ANOVA and a Student’s t-test. Time-to-event measures (disease onset and survival duration) were analyzed using Kaplan–Meier survival fit analysis. The Cox proportional hazards model was used to analyze the effect of RG108 on survival and to determine hazard ratios. There was no censoring of mice due to treatment-related deaths. A one-way ANOVA followed by Tukey post-hoc test was used for statistical comparisons for time-to-event measures. 

## 3. Results

### 3.1. DNMT Enzyme Activity Is Increased in Spinal Cord and Skeletal Muscle of hSOD1 tg Mice 

DNMT enzyme activity was measured in tibialis anterior (TA) and whole spinal cord and from non-tg mice and different genotypes of hSOD1 tg mice at early symptomatic stages of disease (Figure 1A). In non-tg mice, skeletal muscle DNMT enzyme activity was significantly higher (*p* < 0.05) than activity in the spinal cord (Figure 1A). Skeletal muscle DNMT enzyme activity was higher in hSOD1 muscle-specific tg mice with mutant G37R (*p* < 0.01) and G93A (*p* < 0.001) variants but not with WT allele, compared to non-tg controls (Figure 1A). Skeletal muscle DNMT enzyme activity in nonconditional tg mice with hSOD1-G37R was significantly higher (*p* < 0.001) compared to non-tg age-matched controls. In the spinal cord, DNMT activity was significantly higher in hSOD1^mus^-G93A mice (*p* < 0. 01) and -G37R mice (*p* < 0.05) compared to age-matched littermate control mice (Figure 1A). In conditional tg mice with hSOD1^mus^-WT, activity was modestly higher (*p* < 0.05) in spinal cord compared to non-tg mice. Nonconditional hSOD1-G37R mouse spinal cord had increased (*p* < 0.01) DNMT activity compared to controls (Figure 1A). 

### 3.2. Global DNA Methylation Is Increased in Spinal Cord and Skeletal Muscle of hSOD1 tg Mice

Because DNMT enzymatic activity was increased, the levels of the product of this biochemical reaction was measured. Genomic DNA was assayed for global 5mC content in whole spinal cord (Figure 1B) and TA (Figure 1C) of presymptomatic (young) and symptomatic (old) conditional hSOD1^mus^ mice expressing WT, G37R, and G93A genes. Global 5mC levels were similar in spinal cord of all young mouse genotypes (Figure 1B). In older symptomatic mice, global 5mC was increased (*p* < 0.05) in the spinal cord of hSOD1^mus^-G37R, -G93A and -WT mice compared to older non-tg mice (Figure 1B). In non-tg mouse skeletal muscle (Figure 1C), 5mC levels were ~3-fold higher (*p* < 0.01) in young mice compare to older mice. In hSOD1 tg mice, the large difference between young and old mice was greatly diminished. Older symptomatic hSOD1 mice had significantly increased (*p* < 0.05) 5mC levels in skeletal muscle compared to older non-tg mice (Figure 1C). 

### 3.3. 5mC Is Localized to Spinal Cord MNs and Skeletal Muscle Satellite Cells

Because 5mC was increased in total tissue DNA extracts, we next wanted to localize the 5mC to cells. IHC for 5mC has been validated as a good read-out for DNMT activity and in vivo drug and genetic modulation [29,30] and was used to determine if epigenetic changes occur in a severe mouse model of ALS. Significant accumulation of 5mC was seen in spinal MNs in presymptomatic (*p* < 0.001) and early symptomatic (*p* < 0.01) nonconditional high-expressing hSOD1-G93A mice compared to age-matched non-tg littermates (Figure 1D–F). The 5mC accumulation occurred in the nucleus and cytoplasm of MNs and in puncta within dendrites (Figure 1D,E), previously shown to correspond to mitochondrial DNA methylation [29,30].

IF was used to localize 5mC in TA of nonconditional hSOD1-G93A mice. The TA appeared as a mosaic of myofibers positive or negative for hSOD1 with hSOD1-positive small cells seen at the myofiber margins (Figure 1G). Colocalization of hSOD1 with MyoD identified these cells as satellite cells (Figure 1H) that were also enriched in 5mC (Figure 1J). EM identified subsets of satellite cells in presymptomatic mice undergoing apoptosis (Figure 1J), but the number of hSOD1-positive cells that were MyoD- or 5mC-positive did not change significantly with disease stage (Figure 1K).

### 3.4. Gene Promoter CpG Hypermethylation Occurs in Skeletal Muscle of hSOD1 tg Mice

Our 5mC results so far were global assessments that did not relate to specific genes. To refine our 5mC analysis with gene-specific proof-of-principle demonstration of epigenetic changes occurring in one of our ALS mouse models, we did genome-wide promoter CpG methylated-DNA immunoprecipitation (MeDIP)-chip analysis in skeletal muscle of G37R-hSOD1^mus^ mice (Figure 2).

Genes in ALS mouse skeletal muscle (biceps femoris) with significantly increased (>2-fold) hypermethylation (Appendix A) compared to littermate age- and sex-matched non-tg control mice were grouped into functional classes (Figure 2). The predominant differentially methylated gene promoters were distributed in all three promoter classes (high, intermediate, and low CpG density) [73]. Hypermethylated genes (~856 genes) found in biceps femoris of G37R-hSOD1^mus^ mice were grouped into categories of cytoskeletal (15%), intracellular signaling (11%), mitochondrial (10%), plasma membrane receptors-transporters (9%), transcription factors (9%), and nuclear-DNA metabolism-chromatin (8%). Other significantly hypermethylated gene promoters with a smaller representation fell into groups designated as proteostasis (7%), RNA metabolism (6%), and cell death (6%). Representative genes with significantly increased methylation are shown in Appendix A.

### 3.5. DNMT Levels Increase in Spinal Cord and Skeletal Muscle Nuclear Fractions of hSOD1 tg Mice

5mC is marked in the genome by the enzymatic activities of DNMTs. We therefore assessed different DNMT protein levels in different tissue fractions. DNMT1 and DNMT3A proteins were measured in spinal cord nuclear fractions by immunoblotting (Figure 3). 

The mobility in denaturing SDS-PAGE and detection of these proteins were consistent with our previous observations [29,30]. DNMT1 levels were significantly increased 30–50% above control levels in nonconditional hSOD1-G37R (*p* < 0.01), -G93A, (*p* < 0.05) and -WT (*p* < 0.05) mice (Figure 3A,B). DNMT3A was significantly increased (*p* < 0.05) in hSOD1-G93A and -WT mice (Figure 3A,B). In conditional tg mice with skeletal muscle-restricted expression of hSOD1, DNMT1 was significantly increased (*p* < 0.05) in hSOD1^mus^-WT mouse spinal cord nuclear extracts but not in the other tg mouse genotypes (Figure 3C,D). DNMT3A levels were not changed significantly in spinal cord nuclear extracts in any of these tg mouse lines (Figure 3C,D).

DNMT1 and DNMT3A proteins were measured in skeletal muscle nuclear fractions by immunoblotting (Figure 4). 

DNMT1 levels were significantly increased ~40% above (*p* < 0.05) control levels in nonconditional hSOD1-WT mice (Figure 4A,B). In contrast, DNMT3A levels were 3-fold higher (*p* < 0.001) in skeletal muscle nuclear fractions in nonconditional hSOD1-G37R mice, while nonconditional hSOD1-G93A and -WT mice had more modest significant (*p* < 0.05) increases (Figure 4A,B). With skeletal muscle-restricted expression of hSOD1, DNMT1 was unchanged in all three tg mouse genotypes, while DNMT3A was significantly increased in hSOD1^mus^-WT (*p* < 0.01) and hSOD1^mus^-G93A (*p* < 0.05) mice (Figure 4C,D). 

### 3.6. DNMT Levels Increase in Chromatin Bound Fractions of hSOD1 tg Mouse Spinal Cord and Skeletal Muscle 

Within the nucleus, in order for DNMTs to mark DNA with 5mC, the enzymes need to access chromatin. DNMT1 and DNMT3A proteins were measured in spinal cord total chromatin fractions by immunoblotting (Figure 5).

DNMT1 was increased 2.5-fold (*p* < 0.001) in chromatin extracts of spinal cord from nonconditional hSOD1-G93A mice (Figure 5A,B). Chromatin associated DNMT1 was not detected in the G37R and WT genotypes of nonconditional hSOD1 tg mice. In contrast, DNMT3 was significantly increased in chromatin fractions of spinal cords from nonconditional hSOD1-G93A (*p* < 0.01), -G37R (*p* < 0.05), and -WT (*p* < 0.05) mice (Figure 5A,B). In skeletal muscle-restricted hSOD1 expressers, all three genotypes had significantly increased chromatin enrichment of DNMT1 (Figure 5C,D); hSOD1-G37R (*p* < 0.05) and hSOD1-G93A (*p* < 0.01) mice had significantly increased chromatin enrichment of DNMT3A, but hSOD1-WT mice did not (Figure 5C,D). 

DNMT1 and DNMT3A proteins were measured in skeletal muscle chromatin fractions by immunoblotting (Figure 6). 

Nonconditional hSOD1-G37R (*p* < 0.001) and -G93A (*p* < 0.01) had significant ~ 2-fold enrichments of DNMT1 in skeletal muscle chromatin (Figure 6A,B), while hSOD1-WT mice had a lesser increase. A 2.5-fold increase (*p* < 0.001) in chromatin-associated DNMT3A was seen in hSOD1-G37R skeletal muscle, but there was no significant increase in hSOD1-WT mice and hSOD1-G93A mice (Figure 6A,B). In skeletal muscle-restricted expressers, chromatin-associated DNMT1 was increased ~2-fold in hSOD1^mus^-G37R (*p* < 0.01) and hSOD1^mus^-G93A (*p* < 0.01) mice and was ~50% above control (*p* < 0.05) in hSOD1^mus^-WT mice (Figure 6C,D). hSOD1^mus^-WT mice had a significant increase (*p* < 0.05) in chromatin DNMT3A (Figure 6C,D), while hSOD1^mus^-G37R and hSOD1^mus^-G93A mice were not different from controls.

### 3.7. hSOD1 Interacts with DNMT3A in Skeletal Muscle of tg Mice

It is possible that mutant hSOD1 has a gain in function or altered function by aberrantly interacting with DNMTs. IP was used to determine if hSOD1 can directly influence epigenetic regulatory proteins in mouse tissue (Figure 7). 

Co-IP demonstrated that hSOD1-G37R and hSOD1-G93A interacted with DNMT3A in skeletal muscle mitochondrial-enriched fractions (Figure 7A). Co-IP of hSOD1 and DNMT1 was completely negative (Figure 7B). A low level of interaction occurred with hSOD1 and DNMT2 (Figure 7C).

### 3.8. DNA Methylation Is Increased in Human ALS CNS

To determine if our work on DNA methylation in mouse models of ALS has pathological similarities to human ALS, IHC was used to localize 5mC in postmortem brain and spinal cord of patients with ALS and age-matched control individuals (Figure 8). In motor cortex, the number of pyramidal neurons enriched in 5mC was significantly increased (*p* < 0.01) in ALS cases compared to control (Figure 8A,B,H). In the spinal cord, remaining ventral horn lateral group MNs in ALS patients showed prominent somatodendritic attrition and significant (*p* < 0.001) nuclear hypermethylation compared to control individuals (Figure 8C,D,H). Additional observations focused on spinal cord interneurons. Subsets of neurons smaller than attritional MNs showed strong nuclear labeling for 5mC (Figure 8D). These cells could be near-endstage attritional MNs, as described [62], or spinal cord interneurons. Spinal cord interneurons are known to degenerate in tg hSOD1 mouse models of ALS [55,74]. Dual-labeling for 5mC and parvalbumin, an interneuron marker [55], showed that spinal interneurons in control subjects generally had low detectable nuclear 5mC (Figure 8E,I). In ALS patients (Figure 8F,G,I) the proportion of spinal interneurons with nuclear enrichment of 5mC was significantly higher than in controls (*p* < 0.01).

### 3.9. RNA Methylation Is Aberrant in ALS Mouse Spinal Cord and Skeletal Muscle 

Epigenetic pathology could be limited to DNA methylation or it could be broadly manifested to include other nucleic acid such as RNA. We therefore evaluated RNA methylation in hSOD1 tg mice (Figure 9). 

ELISA was used to measure 6mA amounts in whole spinal cord of presymptomatic (6-week-old) nonconditional hSOD1-G93A tg mice and age-matched littermate non-tg mice (Figure 9A). 6mA levels were significantly (*p* < 0.01) reduced in hSOD1-G93A mice compared to control mice (Figure 9A). Whole spinal cord 6mA levels were also measured in presymptomatic (12 months old) conditional hSOD1^mus^-G93A tg mice and age-matched littermate non-tg mice, but these mice, in contrast to the nonconditional mutants, had an 8-fold increase in 6mA levels compared to their controls (Figure 9B).

IHC for 6mA in non-tg mouse spinal cord revealed robust IR associated with MNs (Figure 9C) that was abolished completely by RNase treatment (Figure 9C). Non-tg mouse MNs had a 6mA enrichment pattern resembling Nissl staining of ribonucleoprotein (Figure 9C, non-tg, inset, hatched arrow). The relative enrichments of 6mA IR appeared to decrease progressively in spinal cord of presymptomatic and early symptomatic nonconditional G93A-hSOD1 mice (Figure 9C). The 6mA IR patterns in MNs in these mice were consistent with cytoplasmic vacuolation (Figure 9C, presym and sym, insets, hatched arrows). Some neurons in presymptomatic mice had round intensely positive 6mA cytoplasmic inclusions (Figure 9C, presym, inset, solid arrow). 

Key RNA methylation enzymes in spinal cord and skeletal muscle of hSOD1 ALS mice were measured by Western blotting (Figure 10). 

FTO was detected as a doublet band at ~58–65 kDa in spinal cord (Figure 10A, top); in skeletal muscle, FTO was mostly seen as a single band (Figure 10A, bottom). Reports have shown FTO at this size [75,76]. In nonconditional hSOD1-G93A tg mouse spinal cord and skeletal muscle, FTO was severely reduced (*p* < 0.001) to 30–50% of control at presymptomatic and symptomatic stages of disease (Figure 10D,E). METTL3 was detected as a single protein at ~70 kDa in spinal cord (Figure 10B top and middle), but in skeletal muscle METTL3 was seen as a doublet band (Figure 10B, bottom). Reports have shown METTL3 at this size and as a doublet [76,77]. We detected METTL3 in different subcellular-enriched fractions of spinal cord, including mitochondrial-enriched (Figure 10B, top P2), cytosolic (Figure 10B, top S2), and nuclear-enriched (Figure 10B, top, P1). Within these fractions, METTL3 appeared lowest in the nuclear-enriched fraction in non-tg mice (Figure 10B, top). When spinal cord and skeletal muscle P1 fractions of nonconditional hSOD1-G93A tg mice and age-matched littermate non-tg mice were compared, METTL3 was significantly increased (*p* < 0.01 and *p* < 0.001) in presymptomatic and symptomatic mutant mice (Figure 10D,E). ALKBH5 was detected as ~40 or 50 kDa bands in spinal cord, depending on tissue subcellular fraction (Figure 10C, top and middle). These two isoforms of ALKBH5 have been seen before [76,78], but the tissue fractionation subcellular distribution was not reported. In the mitochondrial-enriched fraction, 40 and 50 kDa ALKBH5 bands were seen (Figure 10C, top). In the cytosolic soluble fraction, only the 40 kDa ALKBH5 band was seen (Figure 10B, top). In the nuclear-enriched fraction, only the 50 kDa band was detected (Figure 10B, top). When spinal cord and skeletal muscle mitochondrial-enriched fractions of nonconditional hSOD1-G93A tg mice and age-matched littermate non-tg mice were compared, ALKBH5 levels were significantly (*p* < 0.05) reduced in presymptomatic mouse spinal cord (Figure 10D) but not in presymptomatic mouse skeletal muscle (Figure 10E).

### 3.10. RNA Methylation Is Depleted Severely in Human ALS CNS

To determine if our work on RNA methylation in mouse models of ALS has pathological similarities to human ALS, IHC was used to localize 6mA in postmortem brain and spinal cord of patients with ALS and age-matched control individuals (Figure 11). 

The number of pyramidal neurons enriched in 6mA decreased significantly (*p* < 0.05) in the motor cortex of ALS cases compared to control (Figure 11A,B,G). Moreover, chromatolytic Betz cells in ALS cases excluded 6mA from the nucleus (Figure 11B inset). In subcortical white matter of motor cortex, the number of cells with an oligodendrocyte nuclear morphology that were positive for m6A was severely reduced (*p* < 0.01) in ALS cases (Figure 11C,D,G). In spinal cord, MNs remaining in ventral horn lateral groups in ALS patients with enrichment of 6mA were significantly reduced (*p* < 0.001) compared to control individuals (Figure 11E–G). For corroboration, ELISA was used to measure global 6mA amounts in whole cervical spinal cord of human ALS and age-matched control cases. 6mA amount was robustly (*p* < 0.001) reduced in human ALS (Figure 11H).

### 3.11. RG108 Delays Disease Onset and Extends the Lifespan of hSOD1-G93A ALS Mice

Because DNA methylation abnormalities were seen in mouse models of ALS and in the real human disease, we returned to the mouse model to examine if this pathology was meaningful as a therapeutic target. RG108 treatment was tolerated well, and it modified several disease parameters. RG108 significantly (*p* < 0.01) delayed disease onset (Figure 12A) as determined by motor activity (vehicle 100 days versus RG108 120 days). 

RG108 appeared to be biologically active in vivo because treatment reduced significantly the number of skeletal muscle satellite cells positive for 5mC (Figure 12B). In addition, RG108 importantly extended maximal survival from 130 days to 160 days (Figure 12C). RG108 was tolerated well based on no observed motor behavioral performance deficits and histological toxicity in liver, heart, and intestine (Appendix A).

## 4. Discussion

ALS appears mechanistically as a heterogeneous disease with familial and sporadic forms. Familial forms are caused by monogenic mutations in several genes, most commonly *c9orf72*, *tdp43*, *sod1*, and *fus* [6]. However, known genetic risk factors identified as ALS-causing do not explain sufficiently the different phenotypes and clinical courses of ALS. Identical twins with familial forms of ALS show disease discordance [8], and the majority of diagnosed ALS cases are sporadic with no Mendelian traits. Age is an important risk factor for ALS [4]. Maleness is also a risk factor for ALS [5]. These factors could allow for a melding of ALS-causing variables with the distinctive epigenetic clock of individuals [79,80,81] and intrinsic and extrinsic factors that influence DNA methylation patterns [82]. Fascinating recent work has shown that disease-discordant monozygotic twins and triplets display differential gene methylation [8], and individuals with underlying *c9orf72* mutations have CpG-island hypermethylation upstream of the *c9orf72* repeat and DNA methylation-age acceleration associated with disease duration and age of onset [83]. Newer work shows in sporadic ALS that DNA methylation age acceleration associates with earlier age of disease onset and shortened lifespan [84]. Environment and lifestyle, such as occupational and leisure exposures and diet, might also affect disease vulnerability and course and can influence epigenetic signatures [85,86]. This idea is exemplified by the finding that Italian soccer players have increased risk for ALS [87,88] that might tie into the neuromuscular junction and its sensitivity to age-related epigenetic modifications [89]. Thus, there is good rationale to study DNA methylation in ALS.

Other DNA methylome studies have focused on human ALS. DNA methylation in sporadic ALS was examined in *sod1* and *vegf* [90], in members of the metallothione gene family [91], and in the astroglial glutamate transporter *eaat2* gene promoter [92]. No differences were found in DNA methylation patterns in sporadic ALS cases compared to control cases. A genome-wide analysis of frontal cortex (Brodmann area 46) DNA methylation in sporadic ALS revealed significant hypermethylation of genes involved in calcium dynamics, oxidative stress, and synapses [93]. Another study of sporadic ALS and control spinal cord DNA identified 112 differentially methylated genes with functions in immune and inflammatory responses [94]. 

In this study, we approached the potential involvement of DNA methylation in ALS pathogenesis by using hSOD1 mouse models and human postmortem CNS. In tg mice, focusing on skeletal muscle and spinal cord, we analyzed DNA methylation at several biological tiers, including global 5mC content and its cellular localization, gene promoter CpG island methylation, DNMT enzyme activity, DNMT protein levels in nuclear and chromatin bound tissue extracts, and DNMT interactions with mutant hSOD1. Our additional work on human ALS focused on the quantitative cellular localizations of 5mC in motor cortex and ventral horn of spinal cord. We make a tentative case for aberrant epigenetic regulation in tg mouse models of ALS and in human ALS. Knowing that DNMTs are already targets for FDA approved clinical cancer treatments [95] and that DNMT inhibition is neuroprotective in an axotomy model of MN death [29], we then did a drug experiment in hSOD1 mice testing the therapeutic efficacy of the highly specific DNMT inhibitor RG108. 

The results on 5mC and DNMTs in hSOD1 tg mice are new and salient for several reasons. DNMTs have not been studied before in tg mouse models of ALS. We studied several different tg hSOD1 mouse lines. The hSOD1^mus^ mice were provocative because hSOD1 presence (WT and mutant variants) only in skeletal muscle induced DNA methylation changes in the spinal cord of mice as they aged. We do not know yet if there is an epigenetic clock-related DNA methylation age acceleration; we assessed mice at only one younger time, and the young tg mice and age-matched non-tg mice were not different. An increase in spinal cord DNMT enzymatic activity in the older mice with hSOD1 expressed only in skeletal muscle is consistent with the increased 5mC content in spinal cord. This finding could mean that there is peripheral retrograde or afferent anterograde control of CNS DNA methylation. hSOD1^mus^ tg mice also had increased 5mC and DNMT enzymatic activity in skeletal muscle. Nonconditional hSOD1-G37R mice had increased DNMT enzymatic activity as well; thus, entirely different hSOD1 tg mouse lines had similar trends. A remarkable anatomical finding was the enrichment of 5mC in skeletal muscle satellite cells of hSOD1^mus^ tg mice, particularly those satellite cells expressing hSOD1. An interesting hypothesis stemming from this finding is that satellite cells in ALS skeletal muscle undergo DNA methylation age acceleration (e.g., they adopt an aberrant fast-paced epigenetic clock). From our EM, skeletal muscle satellite cells of hSOD1^mus^ tg mice undergo apoptosis; perhaps this is programmed by DNA hypermethylation. Previous work has shown that DNMT activation and DNA methylation can drive spinal MN apoptosis in cell culture and in vivo [29].

Our genome-wide gene promoter GpG island methylation experiment provides novel insight on epigenetic changes occurring in skeletal muscle of a hSOD1 tg mouse model of ALS at a presymptomatic stage of disease. We need to verify concordance using gene-specific expression profiling. We used MeDIP and CpG plus promoter arrays. We identified in the biceps femoris of hSOD1^mus^-G37R mice ~856 genes with significant hypermethylation. These genes have diverse functions, those with the most common hypermethylation having functions in the cytoskeleton, intracellular signal transduction, and nucleus. Fewer genes with significant hypermethylation in skeletal muscle were related to mitochondrial biology, proteostasis, RNA homeostasis, and cell death. The cytoskeleton has been implicated in MN diseases for a long time [96,97]. We found *troponin-I* (t*nni1*) hypermethylated. Tnni1 is slow troponin, in contrast to fast troponin that is often implicated in human ALS and is the basis of clinical trials with troponin activators [98]. *Tnni1* loss of function gene mutations cause myopathy [99]. Pathogenically there could be a disease continuum between ALS and primary myopathy. This idea is supported by tg mice expressing hSOD1 only in skeletal muscle [43,44,45,46]. Other hypermethylated genes were related to chromatin modification and cell stress responses. We found *sirtuin6* hypermethylated in hSOD1 tg mice. Sirtuin6 protects against stress-induced DNA damage and apoptosis [100] and stabilizes skeletal muscle metabolism [101]. Sirtuin6 is also cold-inducible and regulates thermogenesis [102]. In this regard, Sirtuin6 is of interest because repetitive periodic whole-body hypothermia can extend the lifespan of ALS mice [103]. hSOD1^mus^-G37R mice also had hypermethylation of *mex3d* gene. Mex3d promotes survival of proliferative cells [104]. Loss of Mex3d could be related to the apoptosis of skeletal muscle satellite cells in hSOD1 mice seen by EM. Functional categories with the least number of hypermethylated gene promoters were endocrine, vascular, trophic-growth factor, and inflammation. Interestingly, only approximately 1% of the hypermethylated genes detected had a functional role in inflammation. Because the mice analyzed were presymptomatic, this could mean that inflammatory changes are secondary to evolving skeletal muscle pathology in ALS. Our identification of the hypermethylation of some endocrine related genes, *neuroexophilin-4* (*nxph4*) and *thyroid hormone receptor interacting protein-12* (*trip12*), is relevant to the emerging concept that ALS is a metabolic syndrome [45,47,48,49]. Nxph4 is a secreted neuropeptide-like glycoprotein and a positive regulator of glycolysis and mitochondrial complex I [105]. Trip12 has many functions, including DNA damage and repair, chromatin remodeling, and muscle fiber selective proteasomal degradation of the multifaceted transcription factor Sox-6 [106,107]. The possible causal roles of Tinn1, Sirtuin6, Mex3d, Nxph4, and Trip12 in ALS mechanisms need to be evaluated.

In human ALS, we found increased numbers of MNs with strong nuclear positivity for 5mC. This occurred in motor cortex layer V and in spinal cord. 5mC IR was detected in MNs of age-matched controls, but it was localized to fine chromatin-like strands in the nucleus. In contrast, the nucleoplasm in ALS MNs was positive throughout the nucleus. MNs with somatodendritic attrition [62] had particularly strong 5mC IR. We have described this before in CNS tissues from different ALS cases [29]. Other apparent neurons in the ventral horn with strong nuclear 5mC IR were seen in human ALS cases but not in controls. We reasoned that these cells were either attritional MNs or spinal interneurons, and, if interneurons, they were not likely to be the calbindin-D28 calcium binding protein-positive Renshaw cells based on location and size [74,108]. Double labeling with the interneuron marker parvalbumin identified these cells as spinal interneurons; moreover, these interneurons had somatodendritic attrition. Our finding is the first description of spinal interneurons undergoing an epigenetic pathology in human ALS. Other investigators have identified loss of spinal interneurons in human ALS autopsy tissue [109] and functional abnormalities in spinal interneurons in living ALS patients [108]. We found that spinal cord parvalbumin interneurons and Renshaw cells degenerate in a hSOD1 mouse model of ALS [55,74]. The results here perhaps hint that spinal interneurons and MNs in ALS undergo a multisynaptic cellular network DNA methylation age acceleration as part of the disease mechanisms. DNA methylation age acceleration has been previously identified as an explanation for disease phenotype variation in human ALS with the proposition that epigenetic biological age, rather than chronological age, is an important determinant of disease phenotype [84]. 

To begin addressing this idea of aberrant DNA methylation as a mechanism of disease in ALS, we did an experiment using a hSOD1 mouse model of ALS and RG108 as a therapeutic test drug. RG108 is a synthetic small molecule, non-nucleoside, direct noncompetitive inhibitor of DNMTs that blocks the DNA-binding pocket [110]. It was developed as an alternative to 5-azacytidine and zebularine that form covalent protein-DNA adducts and have global targeting and dose limiting toxicity in clinical cancer treatment [71]. Though we analyzed by Western blotting DNMT1 and DNMT3A individually, the total biochemical catalytic activity we measured reflected total. We thus chose RG108 as a test drug to broadly inhibit DNMTs because of our experience in using this drug [29], we initially wanted to target total activity, and because DNMT3A-specific inhibitors are just becoming available and require in vivo characterization. Moreover, RG108 can reactivate genes silenced by DNA methylation [110]. RG108 treatment increases the expression of anti-senescence genes, while p53 and p16 expressions are suppressed, in ALS patient-derived bone marrow mesenchymal stromal cells [111]; in addition, RG108 treatment of mice can activate the rejuvenation stemness gene *hes5* [112]. p53 suppression would be an important therapeutic target because it is activated in human ALS [63] and in rodent models of MN degeneration [113,114,115]; furthermore, p53 appears to be a strong driver of adult MN apoptosis [114,115]. Thus, RG108 is a very attractive small molecule for MN disease therapeutics. RG108 appears fundamentally nontoxic in cell culture and in vivo [111,112]. Our prior [29] and current results (Appendix A) support this claim. We showed that intracerebroventricular (ICV) administration of RG108 robustly protected mouse spinal MNs in a model of axotomy-induced apoptosis [29]. Another group showed that ICV administration of RG108 protects spinal MNs in a mouse model of spinal and bulbar muscular atrophy [112]. ICV drug administration has less clinical translatability than systemic drug administration [116]. In our study here, we did systemic RG108 treatment of nonconditional hSOD1-G93A mice. The systemic treatment was novel and afforded exploration of DNA methylation and peripheral versus central target tissues as mechanisms of disease in mouse ALS. RG108 improved motor function, targeted skeletal muscle satellite cells to reduce DNA methylation, and extended survival of ALS mice. This finding strengthens the idea that hSOD1 in skeletal muscle is a driver of pathogenesis in ALS [43,44,45,46] and identifies skeletal muscle DNMTs as disease-modifying therapeutic targets in ALS. 

We also studied RNA methylation in hSOD1 mouse models and human ALS postmortem CNS. This examination has not been done before. We analyzed in hSOD1 mice global 6mA content and the levels of RNA methylation writer and eraser enzymes by Western blotting of spinal cord and skeletal muscle extracts. The human ALS work focused on the quantitative cellular localizations 6mA in motor cortex, subcortical white matter, and ventral horn of spinal cord. In control mice, we showed by IHC the enrichment of 6mA in spinal MNs in a Nissl body-like pattern. This pattern was disrupted in ALS mouse MNs. We also discovered in mice a dramatic age-related loss (~90 depletion) of 6mA in spinal cord of non-tg mice with normal aging (2 months versus 12 months). Compared to corresponding age-matched controls, nonconditional hSOD1-G93A mice showed a 50% depletion of 6mA; in contrast, the restricted-expression of hSOD1-G93A in skeletal muscle significantly rescued mice from the age-related depletion of 6mA in spinal cord. The meaning of this 6mA incongruity in the different hSOD1 mouse models is unclear. The rejuvenation of spinal cord 6mA content by hSOD1 expression only in skeletal muscle could depend on concordant levels of 6mA writers and erasers. Changes in the levels of METTL3, FTO, and ALKBH5 in spinal cord were not evidently explanatory of the change in 6mA. In fact, the 6mA writer METTL3 was significantly upregulated in spinal cord of nonconditional hSOD1-G93A mice. Perhaps this represents compensation, but we do not know the tissue levels of the critical methyl donor cofactor S-adenosyl-methionine that declines in rodent brain ~50% with aging [117,118,119]. 

In human ALS CNS, we observed profound 6mA pathology. By IHC, motor cortical neurons, including Betz cells, and spinal MNs showed loss of 6mA IR. 6mA was mislocalized in MNs in an apparent chromatolytic reaction. Subcortical white matter glial cells positive for 6mA were significantly depleted. By ELISA, there was ~75% depletion of 6mA in spinal cord. Perturbations in RNA metabolism in ALS were identified long ago. A marked loss of single MN total RNA was shown in ALS [50]. Hartmann and colleagues [52] determined ~42% losses of total RNA in cervical and lumbar MNs in ALS cases. Hypoglossal and nucleus ambiguous MNs had a significant loss of mRNA [53]. Base composition analysis of single MNs in ALS cervical spinal cord revealed significant loss of adenine compared to control MNs [120], consistent with our finding of 6mA loss. Contemporary techniques show that ALS cases have severe RNA instability, particularly when TDP-43 pathology is present [121]. It is possible that the 6mA loss is secondary to disease progression and signifies endstage or already emergent disease. The human ALS tissue analysis is at terminal disease with extreme pathology and loss of 6mA. The hSOD1 mice perhaps help with interpretation. The nonconditional hSOD1-G93A mice have aggressive disease with marked ultrastructural pathology seen already at 4 weeks of age [55,56] and presymptomatic depletion of 6mA. The hSOD1^mus^-G93A mouse model has a very slow disease progression and apparent retrograde degeneration of spinal MNs involving p53 [45] and, at a presymptomatic stage of disease, the 6mA concentration in spinal cord is significantly elevated. In this context, accumulation of 6mA might be related to disease causality. We could not examine in mice if the RNA methylation pathology is a therapeutic target because highly specific pharmacological activators or inhibitors are not yet available. More work is warranted to better understand the biology of RNA methylation in ALS.

## 5. Conclusions

DNA and RNA epigenetic methylation anomalies are prominent in hSOD1 mouse models of ALS and human ALS. 5mC and related enzymes were generally increased. 6mA was severely depleted in human ALS. Perturbations can occur in skeletal muscle and CNS. The relationships between this DNA and RNA epigenetic pathology and disease causality remain undiscovered. Abnormalities occur in presymptomatic and symptomatic ALS mice, and they are present in human ALS in postmortem tissue at terminal disease. Thus, the molecular mechanisms sustaining this epigenetic pathology might be persistent during the entire course of disease rendering them meaningful to disease causality and therapeutic targeting. Experiments on ALS mice with presymptomatic initiated treatment of RG108 suggest that aberrant DNA methylation in skeletal muscle and possibly the peripheral nervous system is related to disease onset and progression and lifespan and is a therapeutic target for small molecule inhibitors of DNMTs.

## Figures and Tables

**Figure 1 cells-11-03448-f001:**
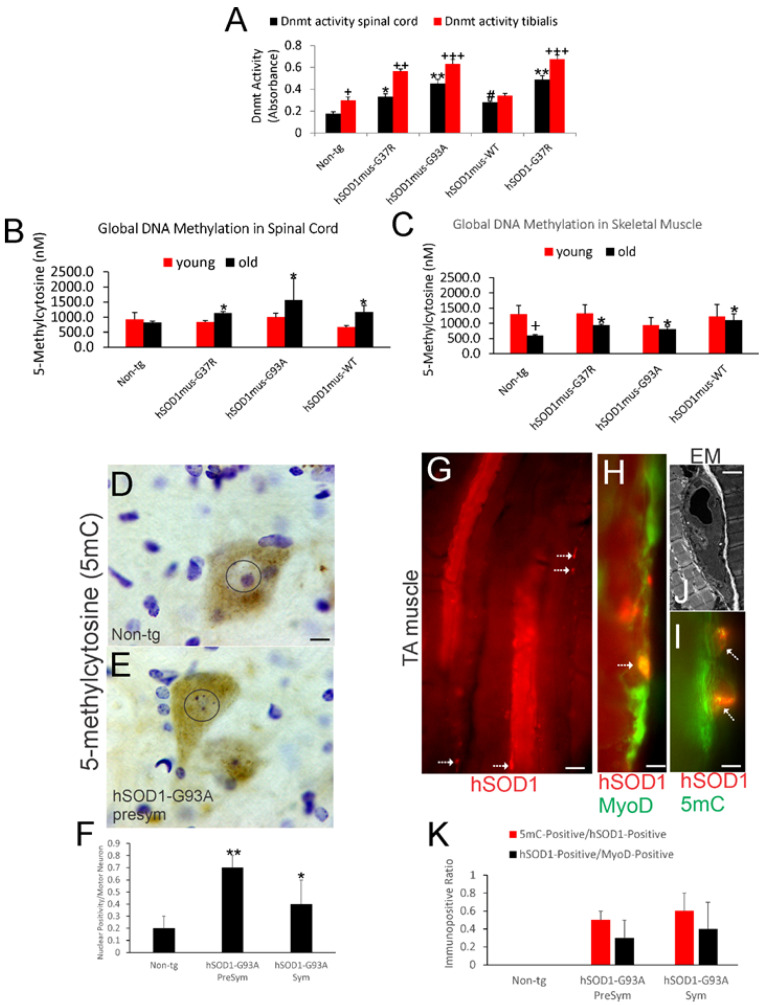
Dnmt enzyme activity and 5mC are increased in mouse ALS. (**A**) Dnmt catalytic activity (mean ± SD, *n *= 6/group) in whole spinal cord and TA muscle in non-tg mice and in different genotypes of hSOD1 tg mice at early symptomatic stages of disease. Non-tg spinal cord vs. non-tg TA muscle, + *p* < 0.05; non-tg vs. tg spinal cord (black bars), # *p* < 0.05, * *p* < 0.01, ** *p* < 0.001; non-tg vs. tg TA muscle (red bars), ++ *p* < 0.01, +++ *p* < 0.001. (**B**,**C**) 5mC levels (nanomolar, nM, mean ± SD, *n* = 6/group) in spinal cord (**B**) and skeletal muscle (**C**) are increased in older (12 months) tg mice compared to age-matched controls but not in young (4–6 months) tg mice. Non-tg vs. tg old mouse spinal cord (**B**), * *p* < 0.05; non-tg young vs. non-tg old skeletal muscle (**C**), + *p* < 0.01; non-tg vs. tg old skeletal muscle (**C**), * *p* < 0.05. (**D**,**E**) IHC shows differences in the localization of 5mC IR (brown staining) in spinal cord MN cell nuclei (enclosed in circles) of non-tg mice (**D**) and hSOD1-G93A tg mice (**E**) at presymptomatic (6 weeks old) stages of disease. (**F**) Counts (mean ± SD, *n* = 6/group) of spinal MNs with full nuclear 5mC IR in hSOD1-G93A mice presymptomatic and symptomatic stages of disease. Non-tg vs. tg, ** *p* < 0.01 and * *p* < 0.05. (**G**) hSOD1-specific IF staining (red) showing the mosaic composition of hSOD1-positive and -negative myofibers in the TA of hSOD1 tg mice. Subsets of small cells on the myofiber margins are positive (hatched arrows). (**H**,**I**) Labeling for MyoD and hSOD1 identifies small cells as satellite cells (H, hatched arrow); subsets are 5mC enriched (**I**, hatched arrow). (**J**) EM shows satellite cells undergo apoptosis in tg mice. (**K**) Counts (mean ± SD, *n* = 6/group) of hSOD1-positive satellite cells, also positive for either MyoD or 5mC, do not change with age in tg mice. Scale bars: (**D**) (same for (**E**)), 7 µm; (**G**), 20 µm; (**H**), 8 µm; (**I**), 5 µm; (**J**), 5 µm.

**Figure 2 cells-11-03448-f002:**
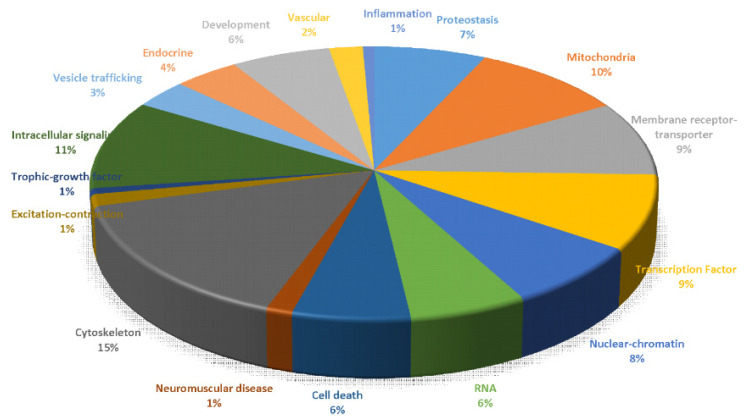
Global promoter CpG island microarray analysis in skeletal muscle (biceps femoris) of presymptomatic (12 months old) hSOD1^mus^-G37R mice compared to age- and-sex-matched non-tg mice. Cut-off for significant hypermethylation in tg mice (*n* = 3) versus age-matched non-tg mice (*n* = 3) was a 2-fold difference. Triage of identified genes with significant hypermethylation into classes was based on known functions of their products. Representative genes assigned to the different categories are shown in Appendix A.

**Figure 3 cells-11-03448-f003:**
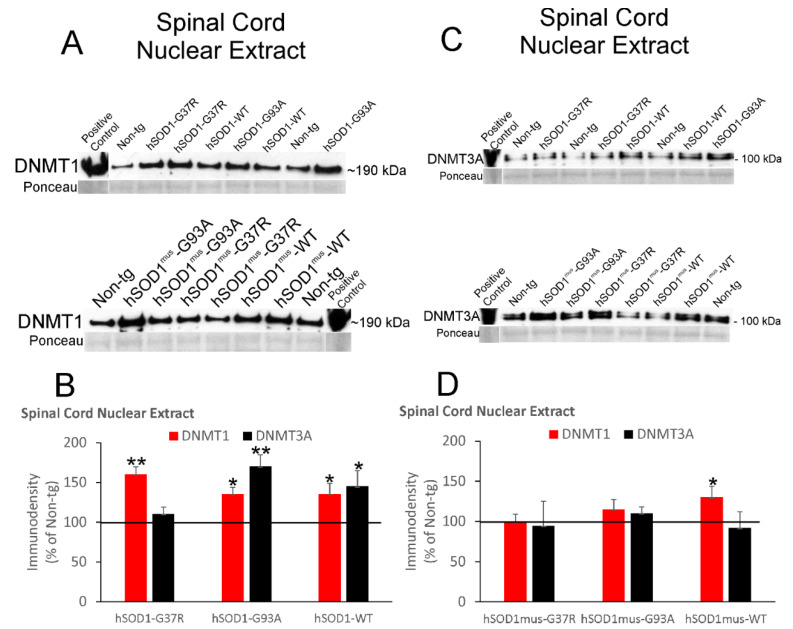
Immunoblotting for DNMT1 and DNMT3A in spinal cord nuclear extracts of nonconditional hSOD1 tg mice and conditional tg mice with skeletal muscle-restricted expression of hSOD1^mus^. Representative blots for DNMT1 (**A**) and DNMT3A (**C**) are shown with different tg mouse lines randomized among lane assignments. Sizes of DNMT1 and DNMT3A (in kDa) are indicated for each blot at right. Each blot had a positive control lane. Ponceau S-stained membranes show protein loading. (**B**) Graph showing IR relative density levels (immunodensity mean ± SD, as % of non-tg mouse IR) for DNMT1 and DNMT3A in nonconditional hSOD1 mouse lines (*n* = 5/genotype). Horizontal line designates non-tg control at 100%. * *p* < 0.05; ** *p* < 0.01. (**D**) Graph showing protein IR relative density levels (mean ± SD, as % of non-tg mouse immunodensity) for DNMT1 and DNMT3A in hSOD1^mus^ mouse lines (*n* = 5/genotype). * *p* < 0.05.

**Figure 4 cells-11-03448-f004:**
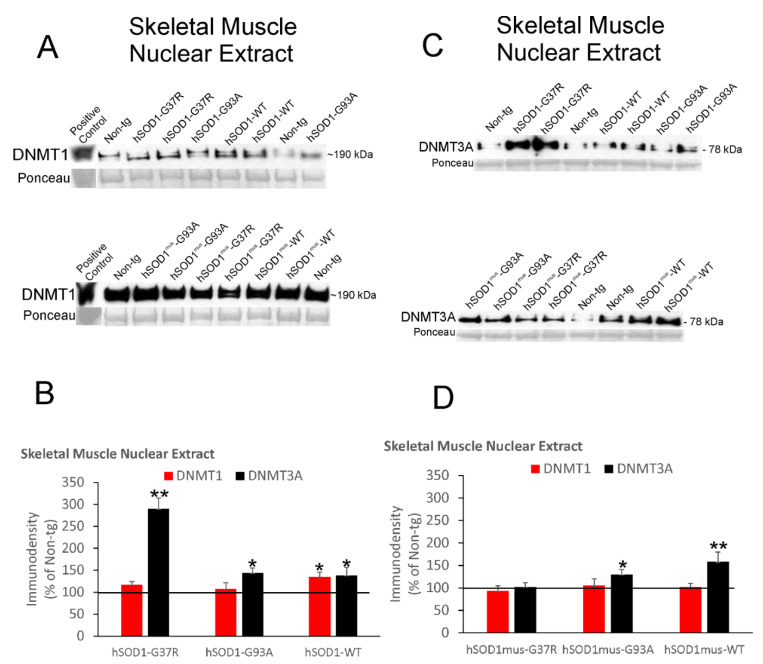
Western blot analysis of DNMT1 and DNMT3A protein levels in nuclear extracts of skeletal muscle from nonconditional hSOD1 tg mice and hSOD1^mus^ mice. Representative immunoblots for DNMT1 (**A**) and DNMT3A (**C**) are shown. The different tg mouse lines were randomized to lane assignments. Sizes of DNMT1 and DNMT3A (in kDa) are indicated for each blot at right. Each blot had a positive control lane. Ponceau S-stained membranes show protein loading. (**B**) Protein IR levels (mean ± SD, as % of non-tg mouse immunodensity) for DNMT1 and DNMT3A in nonconditional hSOD1 mouse lines (*n* = 5/genotype). * *p* < 0.05; ** *p* < 0.001. (**D**) IR protein levels (mean ± SD, as % of non-tg mouse immunodensity) for DNMT1 and DNMT3A in hSOD1^mus^ mouse lines (*n* = 5/genotype). * *p* < 0.05; ** *p* < 0.01.

**Figure 5 cells-11-03448-f005:**
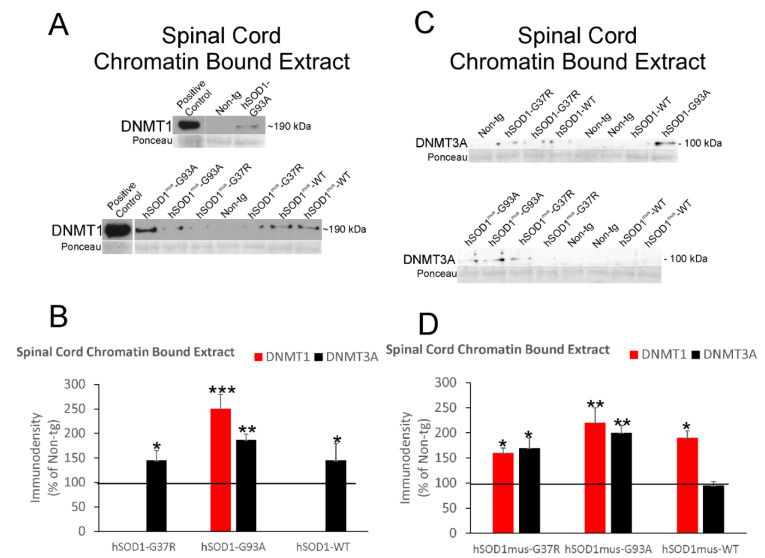
Immunoblotting for DNMT1 and DNMT3A in the chromatin bound extracts of spinal cords from nonconditional hSOD1 tg mice and hSOD1^mus^ mice. Representative blots for DNMT1 (**A**) and DNMT3A (**C**) are shown with different tg mouse lines randomized among the lane assignments. Sizes of DNMT1 and DNMT3A (in kDa) are indicated for each blot at right. Each blot had a positive control lane. Ponceau S-stained membranes show protein loading. (**B**) Protein IR levels (mean ± SD, as % of non-tg mouse immunodensity) for DNMT1 and DNMT3A in nonconditional hSOD1 mouse lines (*n* = 5/genotype). * *p* < 0.05; ** *p* < 0.01; *** *p* < 0.001. (**D**) Protein IR levels (mean ± SD, as % of non-tg mouse immunodensity) for DNMT1 and DNMT3A in hSOD1^mus^ mouse lines (*n* = 5/genotype). * *p* < 0.05; ** *p* < 0.01.

**Figure 6 cells-11-03448-f006:**
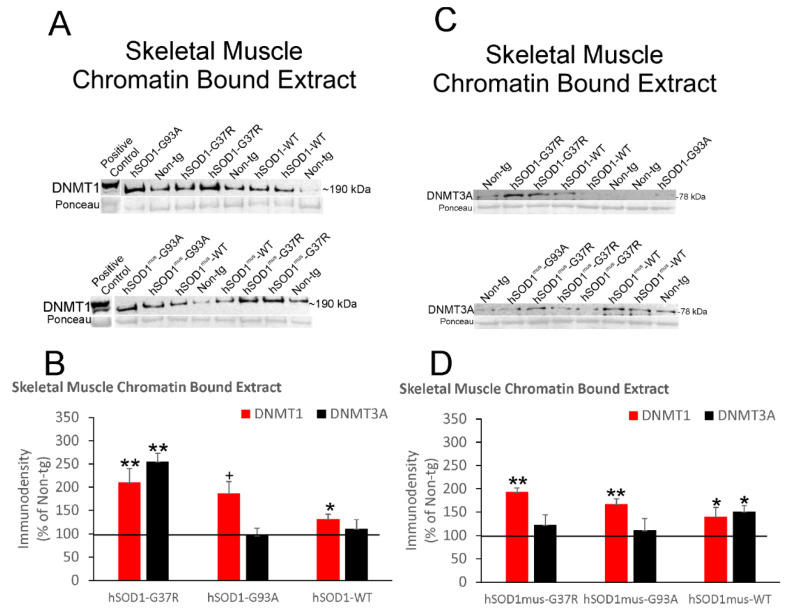
Western blot analysis of DNMT1 and DNMT3A protein levels in the chromatin bound extract of skeletal muscle from nonconditional hSOD1 tg mice and hSOD1^mus^ mice. Representative immunoblots for DNMT1 (**A**) and DNMT3A (**C**) are indicated. Different tg mouse lines were assign to lanes randomly. Sizes of DNMT1 and DNMT3A (in kDa) are indicated for each blot at right. Each blot had a positive control lane. Ponceau S-stained membranes show protein loading. (**B**) Protein IR levels (mean ± SD, as % of non-tg mouse immunodensity) for DNMT1 and DNMT3A in nonconditional hSOD1 mouse lines (*n* = 5/genotype). * *p* < 0.05; + *p* < 0.01; ** *p* < 0.001. (**D**) Protein IR levels (mean ± SD, as % of non-tg mouse immunodensity) for DNMT1 and DNMT3A in hSOD1^mus^ mouse lines (*n* = 5/genotype). * *p* < 0.05; ** *p* < 0.01.

**Figure 7 cells-11-03448-f007:**
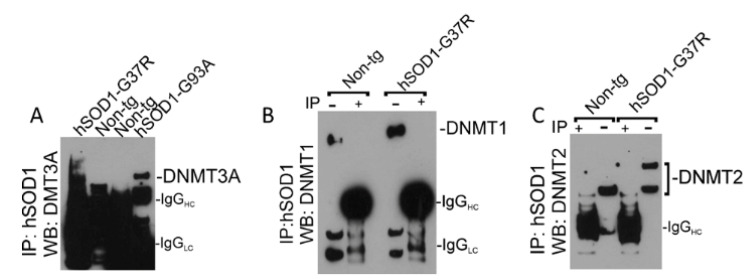
Co-IP identified an interaction between hSOD1 and specific isoforms of DNMTs in skeletal muscle. (**A**) IP of skeletal muscle tissue lysates with human-specific antibody to capture hSOD1 followed by Western blotting (WB) of captured protein for DNMT3A identified co-precipitation of DNMT3A with hSOD1-G37R and hSOD1-G93A but not in skeletal muscle lysates of non-tg mice. Heavy chain IgG (IgG_HC_) and light chain IgG (IgG_LC_) are indicated at ~50 kDa and ~25 kDa, respectively. Very long exposures were required to visualize the DNMT3A bands making the IgG bands overexposed and smudged. (**B**) Co-IP of hSOD1 and DNMT1 was not found, but DNMT1 was detected in skeletal muscle tissue lysate used for input to IP (IP minus lanes). +, denotes hSOD1 antibody was added to the skeletal muscle homogenate. -, denotes hSOD1 was not added to the skeletal muscle homogenate. (**C**) A minor fraction of 60 kDa DNMT2 co-precipitated with hSOD1. DNMT2 was detected in skeletal muscle tissue lysate used for input to IP (IP minus lanes).

**Figure 8 cells-11-03448-f008:**
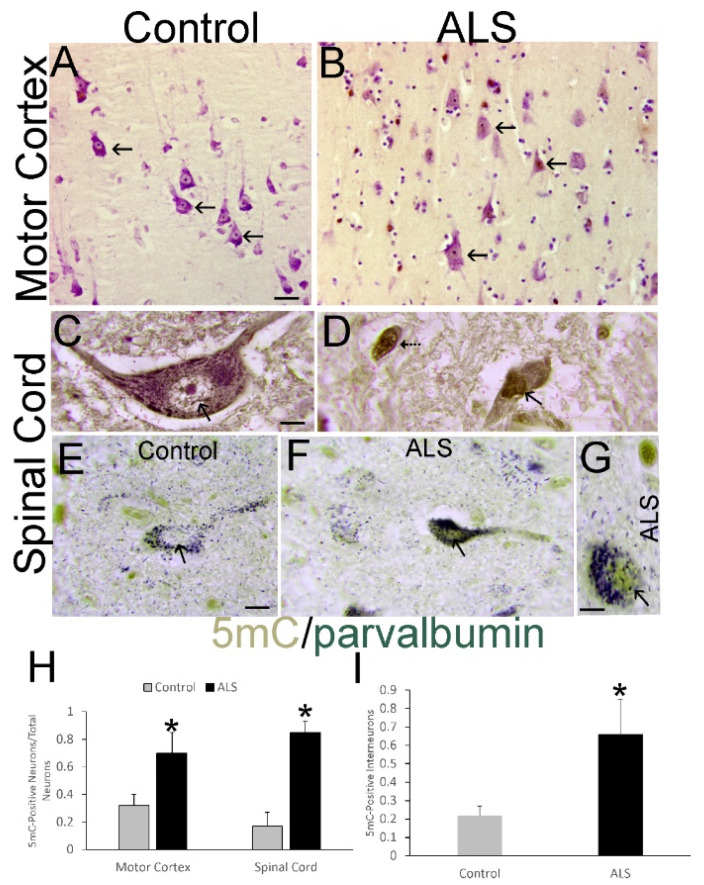
Upper MNs and spinal cord MNs and interneurons undergo hypermethylation in human ALS. (**A**) Layer V pyramidal neurons in control motor cortex (arrows) have low nuclear 5mC IR (brown). (**B**) In ALS motor cortex, numerous layer V pyramidal neurons (arrows) have nuclear 5mC IR (brown). (**C**) Spinal cord MNs in controls have chromatin threads (arrow) decorated with 5mC IR (brown) against a negative nucleoplasm. (**D**) Spinal cord MNs in ALS cases have shrunken nuclei that are strongly positive throughout for 5mC IR (arrow, brown) and cell bodies with somatodendritic attrition. These cells are discernible as attritional MNs because of the marginalized Nissl substance [62]. Other cells (hatched arrow) could be residual MNs or interneurons. (**E**) Interneurons, identified by parvalbumin (arrow, green/black) plus 5mC (brown), show scant DNA methylation IR in controls. (**F,G**) In ALS cases, parvalbumin (green/black, arrow) colocalizes with 5mC (brown). Interneuron in (**G**) is shown at higher magnification. Scale bars: (**A**) (same for (**B**)), 30 µm; (**C**) (same for (**D**)), 10 µm; (**E**) (same for (**F**)), 10 µm; (**G**), 5 µm. (**H**) Counts (mean ± SD) of motor cortical pyramidal neurons and spinal MNs with nuclear IR for 5mC in control (*n* = 6) and ALS (*n* = 8) cases. * *p* < 0.01, motor cortex; * *p* < 0.001, spinal cord. (**I**) Counts (mean ± SD) of parvalbumin-positive spinal cord interneurons with nuclear IR for 5mC in control (*n* = 6) and ALS (*n* = 8) cases. * *p* < 0.01.

**Figure 9 cells-11-03448-f009:**
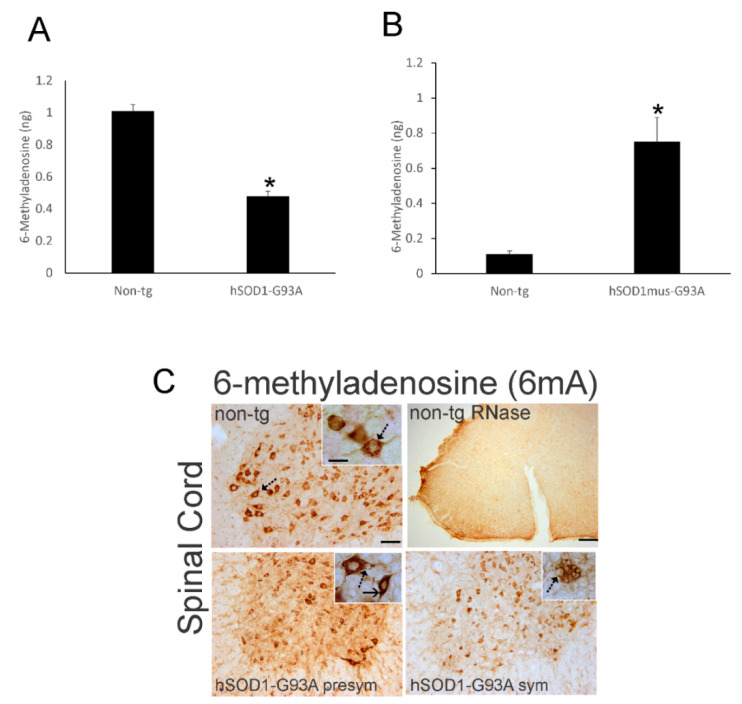
6mA RNA methylation is abnormal in hSOD1 mouse ALS. (**A**) ELISA for m6A content (nanogram [ng], mean ± SD, *n* = 6/group) in whole spinal cord of presymptomatic (6 weeks old) nonconditional hSOD1-G93A tg mice and age-matched littermate non-tg control mice. * *p* < 0.001. (**B**) m6A ELISA (mean ± SD, *n* = 6/group) for whole spinal cord of presymptomatic (12 months old) conditional hSOD1-G93A^mus^ tg mice and age-matched littermate non-tg control mice. * *p* < 0.001. (**C**) IHC for 6mA (brown staining) in lumbar spinal cord ventral horn of presymptomatic (6 weeks old) and symptomatic (12 weeks old) nonconditional hSOD1-G93A tg mice and non-tg control mice. 6mA is enriched in MNs (non-tg, hatched arrow). Insets show MNs (arrows) at higher magnification. Mutant hSOD1-G93A mouse MNs have cytoplasmic vacuoles (hatched arrows) or round strongly positive cytoplasmic 6mA inclusions (solid arrow). For a negative control, spinal cord sections were pretreated with RNase that abolished the IR for 6mA. Non-tg mouse sections were use for RNase treatment because they had the strongest IR. Scale bars: (**C**) non-tg (same for presym and sym), 40 µm; inset (non-tg, same for others), 10 µm; (**C**) non-tg RNase, 100 µm.

**Figure 10 cells-11-03448-f010:**
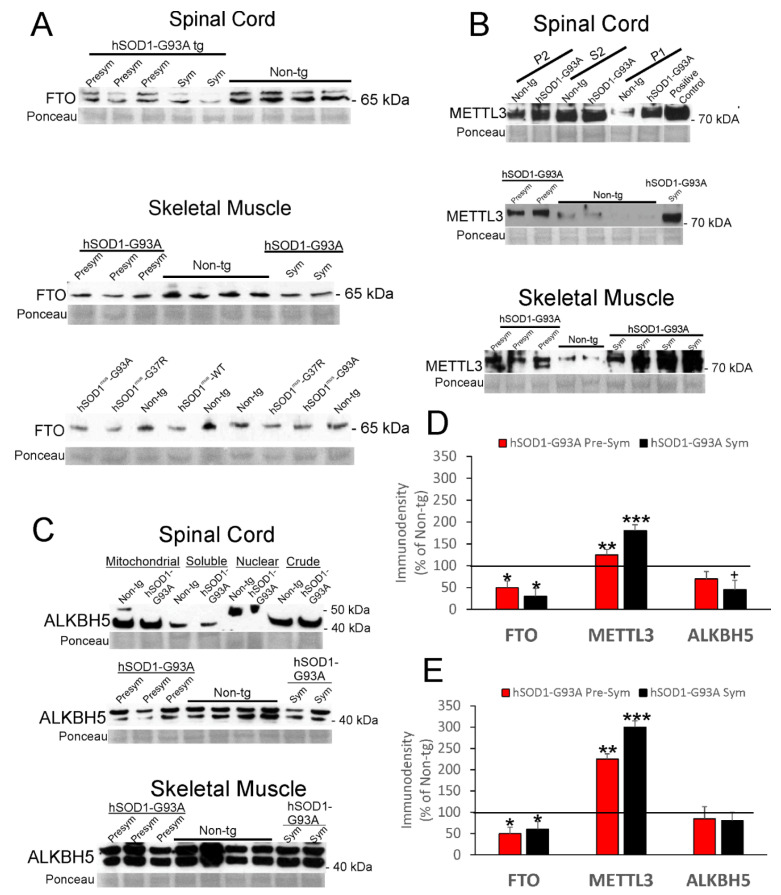
Western blot analysis of RNA methylation-related enzymes in spinal cord and skeletal muscle of hSOD1 tg mice. (**A**) FTO levels in spinal cord ((**A**), top) and skeletal muscle ((**A**), bottom) of nonconditional hSOD1 tg mice (presymptomatic and symptomatic) and conditional hSOD1^mus^ tg mice. Age-matched non-tg littermates were controls. The several different hSOD1 tg mouse lines are shown randomized among the lane assignments. FTO size (in kDa) is indicated for each blot. Ponceau S-stained membranes show protein loading. (**B**) METTL3 levels in spinal cord ((**B**), top and middle) and skeletal muscle ((**B**), bottom) of nonconditional hSOD1 tg mice (presymptomatic and symptomatic) and conditional hSOD1^mus^ tg mice. Age-matched non-tg littermates were controls. METTL3 size (in kDa) is indicated. In top blot, different subcellular fractions of spinal cord were used to assess the presence of METTL3 in mitochondrial-enriched (P2), soluble (S2), and nuclear-enriched (P1) tissue compartments. The middle blot was done using nuclear fractions of spinal cord. The lowest blot in B was done using nuclear fractions of skeletal muscle. (**C**) ALKBH5 levels in spinal cord ((**C**), top and middle) and skeletal muscle ((**C**), bottom) of nonconditional hSOD1 tg mice (presymptomatic and symptomatic) and conditional hSOD1^mus^ tg mice. Age-matched non-tg littermates were controls. ALKBH5 size (in kDa) is indicated. In top blot, different subcellular fractions of spinal cord were used to assess ALKBH5 presence in the mitochondrial-enriched, soluble, and nuclear-enriched compartments and crude homogenate. The middle blot was done using mitochondrial fractions of spinal cord. The lowest blot in C was done using mitochondrial fractions of skeletal muscle. (**D**) Protein IR levels (mean ± SD, as % of non-tg mouse immunodensity) for FTO, METTL3, and ALKBH5 in spinal cord of nonconditional hSOD1-G93A mice at presymptomatic and symptomatic stages of disease (*n* = 6/group). * *p* < 0.001; + *p* < 0.05; ** *p* < 0.01; *** *p* < 0.001. (**E**) Protein IR levels (mean ± SD, as % of non-tg mouse immunodensity) for FTO, METTL3, and ALKBH5 in skeletal muscle of nonconditional hSOD1-G93A mice at presymptomatic and symptomatic stages of disease (*n* = 6/group). * *p* < 0.001; ** *p* < 0.01; *** *p* < 0.001.

**Figure 11 cells-11-03448-f011:**
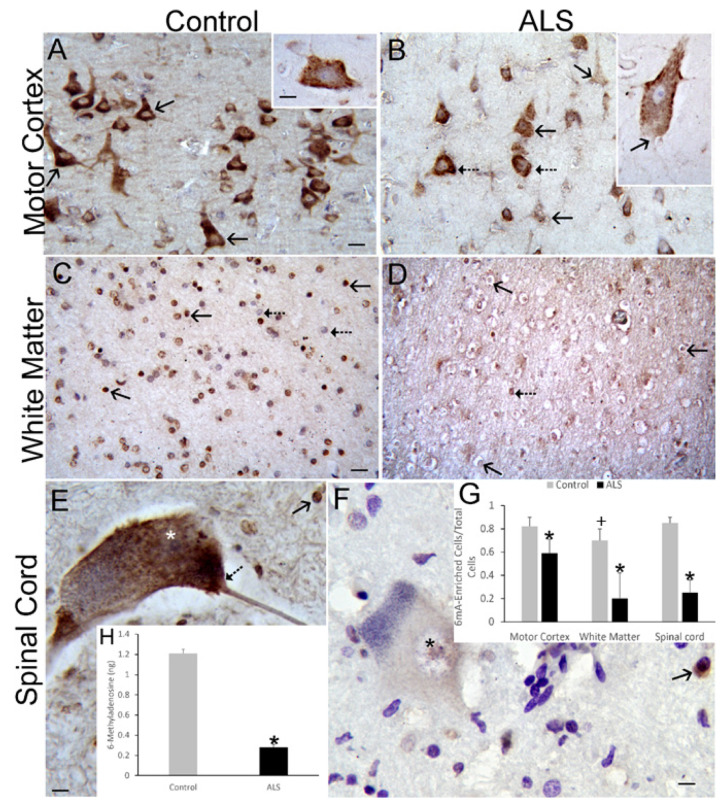
6mA RNA methylation is depleted in postmortem human ALS upper MNs, white matter, and lower MNs. Sections were counterstained with cresyl violet. (**A**) In control motor cortex, layer V pyramidal neurons (arrows) are enriched in 6mA IR (brown). The cytoplasm is strongly positive but the nucleus is less enriched (inset). See G for quantification. (**B**) In ALS motor cortex, layer V pyramidal neurons (solid arrows) have diminished 6mA IR (brown) in the cytoplasm and the nucleus compared to control. Subsets of pyramidal neurons remain enriched (hatched arrows). Neocortical layer V Betz cells, with a chromatolytic morphology (inset), have depleted 6mA in the nucleus, cytoplasm, and axon hillock (arrow). (**C**) Subcortical white matter in human controls is enriched in cells with nuclear IR for 6mA (solid black arrows). These cells have a nuclear morphology of oligodendrocytes [67]. Some cells are less intensely positive (hatched black arrows). (**D**) In the subcortical white matter of human ALS cases, many cells with an oligodendrocyte morphology show 6mA depletion (solid black arrow); few cells appear less affected (hatched arrow). (**E**) In human control spinal cord, the α-MNs are enriched in 6mA throughout the cytoplasm, including the axon hillock (hatched arrow), and nucleus (*). Small glial cells are also positive (solid arrow). (**F**) In human ALS spinal cord, the MNs are attritional and have severe depletion of cytoplasmic and nuclear (asterisk) 6mA. Scale bars: (**A**) (same for (**B**)), 15 µm; (**A**) (inset), 7 µm; (**C**) (same for (**D**)), 20 µm; (**E**) (same for (**F**)), 10 µm. (**G**) Counts (mean ± SD) of motor cortical pyramidal neurons, white matter glial cells, and spinal MNs with enrichment of 6mA in control (*n* = 6) and ALS (*n* = 8) cases. + *p* < 0.05; * *p* < 0.01. (**H**) ELISA determination of the nanogram (ng) amount (mean ± SD) of 6mA in cervical spinal cord of human control (*n* = 6) and ALS (*n* = 8) cases. * *p* < 0.001.

**Figure 12 cells-11-03448-f012:**
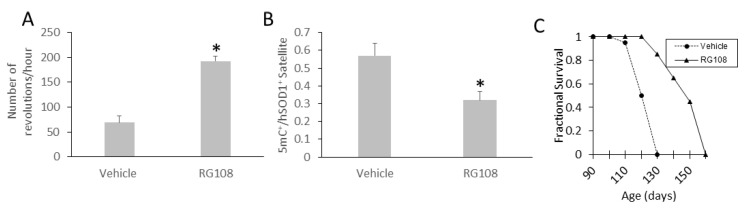
The DNMT inhibitor RG108 is therapeutic in nonconditional hSOD1-G93A mice. RG108 was given ip 2 mg/kg/day starting at 6 weeks of age. (**A**) RG108 restored significant (* *p* < 0.001) motor function (mean ± SEM) on a running wheel in 12-week-old mice (*n* = 10/group). (**B**) RG108 significantly reduced (* *p* < 0.01) the accumulation of skeletal muscle satellite cells positive for 5mC (mean ± SD, *n* = 10/group). (**C**) RG108 treatment of hSOD1-G93A mice (*n* = 20/group) delayed disease onset (vehicle 100 day versus RG108 120 days) and extended maximal survival from 130 days to 160 days.

**Table 1 cells-11-03448-t001:** Primary Antibodies Used.

Target Protein	Commercial Source	Assay
Human SOD1 (human-specific, not reactive with mouse SOD1)	MBL International, clone1G2	Immunoprecipitation (IP), IF
MyoD	BD Pharmingen, clone MoAb5.8A	IF
5mC	Calbiochem, clone 16233D3	IF, IHC
6mA	Proteintech, clone 1D5E10	IHC
DNMT1	Enzo, clone 60B12220.1	Western blotting (WB)
DNMT3A	Enzo, clone 64B1446	WB
DNMT2/TRDMT1	LSBio, rabbit polyclonal	WB
FTO	LSBio, rabbit polyclonal	WB
METTL3	Bethyl Labs, rabbit polyclonal	WB
ALKBH5	Proteintech, rabbit polyclonal	WB
Parvalbumin	Sigma, clone PARV-19	IHC

## Data Availability

Data and materials supporting the conclusions of this work are included herein. L.J.M. is to be contacted to request availability of these materials.

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
