# Peer review of "Aberrant DNA and RNA Methylation Occur in Spinal Cord and Skeletal Muscle of Human SOD1 Mouse Models of ALS and in Human ALS: Targeting DNA Methylation Is Therapeutic"

_cells, 2022, doi:10.3390/cells11213448_

Round 1

Reviewer 1 Report

The authors used different human superoxide dismutase-1 (hSOD1) transgenic (tg) mouse lines, mice expressing non conditionally wildtype (WT) and the G93A and G37R mutant variants, and mice expressing conditionally skeletal muscle-restricted WT and G93A and G37R mutated forms, to demonstrated that hSOD1 mutant mice showed an increased DNA methyl- 21 transferase enzyme activity in spinal cord and skeletal muscle and increased 5-methylcytosine 22 (5mC) levels. Moreover, genome-wide promoter CpG DNA methylation profiling in skeletal muscle of ALS mice identified multiple gene categories with hypermethylation. 5mC accumulated in spinal cord motor neurons and skeletal muscle satellite cells in mice. In addition, the authors found a significant increase in DNA methyltrans- 25 ferase-1 (DNMT1) and DNA methyltransferase-3A (DNMT3A) levels in spinal cord nuclear and chromatin bound extracts of nonconditional hSOD1-G37R, -G93A, and -WT mice com- 27 pared to age-matched non-tg mice.

The authors found that mice with skeletal muscle-restricted expression of hSOD1-G93A, -G37R, and -WT had elevated DNMT1 in chromatin bound fractions, DNMT3A was elevated in nuclear and chromatin bound extracts of skeletal muscle in some hSOD1 variants, mutant hSOD1 interacted with DNMT3A. 6-methyladenosine (m6A) RNA methylation was markedly increased or decreased in mouse spinal cord depending on hSOD1-G93A model, whereas fat mass and obesity as- sociated protein was depleted, and methyltransferase-like protein 3 was increased in spinal cord and skeletal muscle.

Finally, the human ALS tissues showed an increased numbers of motor neurons and interneurons with nuclear 5mC; m6A was severely depleted. Of note, the treatment of hSOD1-G93A mice with DNMT inhibitor improved motor function and extended lifespan by 25%. Overall findings suggest that DNA and RNA epigenetic anomalies are crucial in mouse and human ALS and they could be a potential therapeutic targets for the disease.

The study provides new insights in the field of ALS. However, some experiments and revisions are needed to improve the manuscript before publication.

Major revision:

-          The authors should revise the abstract because it is difficult to be understood.

-          In the results section, figure 1 is missing.

-          The authors should revise the figure 3, 4, 5, 6, 7, 8, 9 because they are too small and difficult to read. Moreover, 12 figures are too much; some of them could be put together.

-          In general, the manuscript is too long and difficult to read, and should be improved.

Author Response

Dear Editors and Reviewers:

Thank you for reviewing our manuscript entitled “Aberrant DNA and RNA methylation occur in spinal cord and skeletal muscle of human SOD1 mouse models of ALS and in human ALS: targeting DNA methylation is therapeutic” for publication consideration in Cells.  We were very pleased with the reviewers’ comments because they were all very constructive and helpful. We have made major revisions to our manuscript and have addressed each of the reviewers’ comments, and, in doing so, the manuscript has been improved and strengthened considerably.

Our responses and corresponding changes to the manuscript in accordance each of the reviewers’ suggestions are identified on a point-by-point basis below. The changes made to the manuscript are RED TRACKED.

Reviewer 1

Rating Comments. Extensive editing of English language and style required.

Author’s response: We have extensively edited the manuscript. These edits include shortening the text, grammar corrections, adding a missing figure, enlarging all the figures, clarifying results, and adjusting conclusions.

                Major revision comments.

  1. The authors should revise the abstract.

Author’s response. The abstract has been revised by identifying key questions, better organizing the questions, deleting long cumbersome sentences, and editing the conclusion.

  1. Figure 1 is missing.

Author’s response. The missing figure is added. We apologize for this oversight.

  1. Figures are too small.

Author’s response. All of the figures have been enlarged. Because the figures have been enlarged, many are nearly full-page or ¾ page in size. This made it difficult to combine figures to reduce the total number of figures.

  1. The manuscript is too long.

Author's response. We have shortened the text to reduce length. Text deletions have been made in the Abstract, Introduction, and Discussion.

Reviewer 2 Report

In this work, Martin and colleagues evaluated the effect of aberrant 5mC and 6mA methylation on rats.  The results were complemented with an evaluation of similar parameters on human biopsies.  The authors showed an abnormal methylation profile in ALS rats that impacted their survival, which was reversed with pharmacological intervention using RG108 (inespecific DNMT inhibitor).  Overall the manuscript contains interesting results, supported by solid data and conveys an interesting study that ultimately will benefit ALS patients in the future.  However, I think there are some issued with the manuscript that need to be addressed before publication.  I suggest returning the manuscript to the authors with the following Major Revisions:

1. I could not find figure 1 in the source material provided.  Please add this to the manuscript.

2. Regarding the graphics of band intensity quantification, I suggest the following improvements:

2.1. Consider improving their readability and contrast for better readbility;

2.2. Consider using the same scale for all of them, of at least use the same for those compiled in the same figure/panel;

2.3. Consider adding bars for the control rat samples analyzed (I assume they will all be of 100%) for better understanding of the results.  I believe this helps to better understand the statistical changes found.

3. For image 5, consider increasing the image/font size for better readability;

4. Regarding Figure 7, consider improving this set of results:

4.1. The photos of the blots have big blurs and band identification is difficult.  Consider providing new images.

4.2. I believe this is an interesting study but not explained fully in the text.  Consider adding context to this set of data and highlighting better the main results and conclusions obtained.  For example, the + and – signs in the gels relate to what experimental approaches?

5. Overall I believe there is not enough link between the different sections, especially between the different results section.  I believe this might hamper the understanding of this work.  Please address this.

6. I feel the data provided for sections 3.8 and 3.10, concerning human samples, could be presented in the beginning to justify the DNMT study performed in rats.  Consider moving these sections to the beginning of the manuscript and providing a link between the two studies.

7. For the IHC images provided, consider:

7.1. Improving their formatting and readability (e.g. increase the zoom of image 9C; image disposition in Figure 8; color caption in Figure 8);

7.2. Providing images with higher magnification for better sub-cellular structure identification;

7.3. Present the associated graphics in full size instead of small figures, as was depicted in Figure 11;

8. Why the authors chose a conditional (hSOD1-G93Amus) and an unconditional (hSOD1-G93A) models for RNase treatment analysis of spinal cord samples instead of 2 corresponding unconditional or conditional models?

9. In figure 11, what was the co-stain used to identify the neurons?

10. Consider expanding the discussion: why were so many specific DNMT enzymes analyzed but only a inespecific DNMT inhibitor was used?  What is the main novelty of your work compared to similar studies?

11. Consider providing objective and/or quantitative data/statements to improve your conclusions section, as I believe this needs to be improved.

Author Response

Dear Editors and Reviewers:

Thank you for reviewing our manuscript entitled “Aberrant DNA and RNA methylation occur in spinal cord and skeletal muscle of human SOD1 mouse models of ALS and in human ALS: targeting DNA methylation is therapeutic” for publication consideration in Cells.  We were very pleased with the reviewers’ comments because they were all very constructive and helpful. We have made major revisions to our manuscript and have addressed each of the reviewers’ comments, and, in doing so, the manuscript has been improved and strengthened considerably.

Our responses and corresponding changes to the manuscript in accordance each of the reviewers’ suggestions are identified on a point-by-point basis below. The changes made to the manuscript are RED TRACKED.

Reviewer 2

Rating Comments. English language are fine/minor spell check required. Results need clearer presentation.

Author’s response: We have edited the manuscript for grammar and spelling. We have provided transitional explanations between the different sections in the results.

Comments & suggestions.

  1. Could not find figure 1.

Author’s response. The missing Figure 1 has been added.

  1. Band intensity quantification.
    • Improve readability

Author’s response. The figures have been enlarged to improve readability.

  • Using the same scale.

Author’s response. All of the graphs in the same figure now have the same scale. Thank you for pointing this out.

  • Consider adding bars for the control samples.

Author’s response. The non-tg control optical densities were set at 100%, so to make more evident the changes, we added a horizontal line across the graph at 100%. This helps to appreciate the differences in the transgenic mice. Adding addition bars to the graphs would increase figure size, which is already a problem, so we thought that this solution would work.

  1. Image 5.

Author’s response. Image 5 size has been increased.

  1. Figure 7.

4.1. The blot has blurs. 

Author’s response. Figure 7 shows the immunoprecipitation experiment results. The black smudging is difficult to avoid. For the detection of hSOD1-DNMT3A interactions we needed to use a lot of immunoprecipitating antibody and long exposures to detect DNMT3A. Consequently, the immunoglobulin bands are very overexposed and dark. We have explained this in Figure 7 legend. We feel strongly that the results are valid and reproducible, but the experiment is difficult, and the blots are inelegant.

  • Figure 7. Explain the + and – signs better.

Author’s response. We have explained the + and – signs in the figure legend better.

  1. The links between the different result sections could be better.

Author’s response. At the beginning of each new section in the results, we have added a transitional statement to help explain why the next set of experiments was done.

  1. The human data could be presented at the beginning of the results.

Author’s response. We tried rearranging the results by having human data first. Because we are very interested in skeletal muscle in ALS, the disadvantage of the having the human data first is that we have no experiments with human skeletal muscle, and there is always the necessity to carefully control for and understand postmortem artifact. Moreover, postmortem skeletal muscle tissue was not available to us. Including skeletal muscle data at the beginning was very important to us. We thought the flow of information was appropriate from 1) epigenetic pathology identification in established mouse models of ALS using optimally prepared tissue samples; 2) using human ALS and control CNS tissues to show that changes seen in the mouse models happen in the human disease; and 3) lastly, that we can manipulate these disease-relevant, reliable, and consistent changes in a mouse model of ALS to therapeutic ends.

  1. IHC images.

7.1. Improve formatting and readability of Fig. 9C; image disposition in Figure 8.

Author’s response. We have increased the sizes of Figure 9 and Figure 8.

  • Need for higher magnification images.

Author’s response.  For Figure 9C we have added insets showing the positive motor neurons at higher magnificent.

  • Figure 11 size.

Author’s response. For Figure 11 we have increased the sizes of the individual graphs and have increased the size of the overall figure.

  1. RNase treatment in Figure 9.

Author’s response. We showed a non-tg mouse spinal cord section treated with RNase as a negative control for the 6mA antibody staining because the non-tg mice show the strongest staining. Showing that this staining could be completely digested was thought to be important. This reasoning was included in the figure legend of Figure 9.

  1. Co-stain in Figure 11?

Author’s response. We have noted in the legend that the co-stain in the figure was cresyl violet.

  1. Consider expanding the discussion of why were so many DNMTs studied but a non-specific inhibitor was used.

Author’s response. We have added brief sentences in the discussion to explain this.

  1. Improve the conclusions section.

Author’s response. We have revised the conclusion section.

Round 2

Reviewer 1 Report

the authors answered all comments

Reviewer 2 Report

The authors have addressed all the issued presented.  I recommend the manuscript to be accepted.